# *Cathepsin D* Plays a Vital Role in *Macrobrachium nipponense* of Ovary Maturation: Identification, Characterization, and Function Analysis

**DOI:** 10.3390/genes13081495

**Published:** 2022-08-21

**Authors:** Dan Cheng, Wenyi Zhang, Sufei Jiang, Yiwei Xiong, Shubo Jin, Fangyan Pan, Junpeng Zhu, Yongsheng Gong, Yan Wu, Hui Qiao, Hongtuo Fu

**Affiliations:** 1Wuxi Fisheries College, Nanjing Agricultural University, Wuxi 214081, China; 2Key Laboratory of Freshwater Fisheries and Germplasm Resources Utilization, Ministry of Agriculture and Rural Affairs, Freshwater Fisheries Research Center, Chinese Academy of Fishery Sciences, Wuxi 214081, China; 3National Demonstration Center for Experimental Fisheries Science Education, Shanghai Ocean University, Shanghai 201306, China

**Keywords:** *cathepsin D*, RNAi, *Macrobrachium nipponense*, ovary maturation, mRNA expression

## Abstract

The oriental river prawn *Macrobrachium nipponense* is an economically important aquacultural species. However, its aquaculture is negatively impacted by the rapid sexual maturation of female *M. nipponense*. The fast sexual maturation produces a large number of offspring which leads to a reduction in resilience, a low survival rate, and an increased risk of hypoxia, this in turn, seriously affects the economic benefits of prawn farming. Cathepsin D is a lysosomal protease involved in the ovarian maturation of *M. nipponense*. In the current study, the cDNA of the gene encoding cathepsin D (*Mn-CTSD*) was cloned from *M. nipponense.* The total length was 2391 bp and consisted of an open reading frame (ORF) of 1158 bp encoding 385 amino acids. Sequence analysis confirmed the presence of conserved N-glycosylation sites and characteristic sequences of nondigestive cathepsin D. The qPCR analysis indicated that *Mn-CTSD* was highly expressed in all tissues tested, most significantly in the ovaries, whereas in situ hybridization showed that expression occurred mainly in oocyte nuclei. Analysis of its expression during development showed that *Mn-CTSD* peaked during the O-IV stage of ovarian maturation. For the RNAi interference experiment, female *M. nipponense* specimens in the ovary stage I were selected. Injection of *Mn-CTSD* double-stranded (ds)RNA into female *M. nipponense* decreased the expression of *Mn-CTSD* in the ovaries, such that the Gonad Somatic Index (GSI) of the experimental group was significantly lower than that of the control group (1.79% versus 4.57%; *p* < 0.05). Ovary development reached the O-III stage in 80% of the control group, compared with 0% in the experimental group. These results suggest that *Mn-CTSD* dsRNA inhibits ovarian maturation in *M. nipponense*, highlighting its important role in ovarian maturation in this species and suggesting an approach to controlling ovarian maturation during *M. nipponense* aquaculture.

## 1. Introduction

*Macrobrachium nipponense*, also known as the oriental river prawn, is an important freshwater aquaculture prawn in China [1]. The development of *M. nipponense* aquaculture has been challenged by the rapid sexual ripening of this species [2], particularly during the breeding season. The ovaries of female prawns mature quickly, which results in various issues, such as the coexistence of multiple generations, high breeding density, and water hypoxia, which significantly impact the aquaculture of this species [3]. Therefore, research has focused on whether it might be possible to slow sexual ripening in this species. Determining the molecular mechanism of ovarian maturation could provide an approach for addressing this issue. 

Previous research focused on determining the molecular mechanism of ovarian maturation in *M. nipponense*, with comparative analyses of the transcriptome of adult female ovaries from the O-I to O-V stages of development [1,4,5]. The enriched KEGG pathways were sorted according to their q-values, with expression changes in the ‘Lysosome’ pathway found to be significantly different. In addition, the expression of a cathepsin was found to be involved in *M. nipponense* ovarian maturation [5]. 

Cathepsin D is an acid lysozyme that hydrolyzes intracellular proteins and peptides to maintain cell homeostasis [6,7,8]. It is involved in the activation of proenzymes and growth factors, brain antigen presentation, epidermal differentiation, and apoptosis in humans and mammals [9]. Research on cathepsin D in aquatic animals has mainly focused on bony fish, showing that cathepsin D is associated, to some extent, with the immune response [7,10,11,12,13]. In addition, cathepsin D is related to immune function in crustaceans, including *Procambarus clarkii* [14], *Fenneropenaeus chinensis* [15], and *Penaeus japonicus* [16]. However, less is known about its role in reproduction in Crustacea. 

Therefore, the present study aimed to characterize the structure of cathepsin D of *M. nipponense* and study its expression pattern and subcellular localization. Furthermore, we investigate its function in ovarian maturation. This study provides insight into the molecular mechanisms involved in ovarian maturation and could be useful for addressing the rapid female sexual maturation of this species.

## 2. Materials and Methods

### 2.1. Experimental Animals

No endangered or protected species were involved in this study. All experimental protocols were approved in October 2021 (Authorization NO. 20211001006) by the Animal Care and Use Ethics Committee in the Freshwater Fisheries Research Center (Wuxi, China).

A total of 120 adult female *M. nipponense* (BW ± SD:0.58 ± 0.16 g) were used in the study and were supplied by the Freshwater Fisheries Research Center, Chinese Academy of Fishery Sciences (Wuxi, China). All the prawns were acclimatized in a circulating water tank for 1 week. The prawns were then euthanized and tissues including the eyestalk, cerebral ganglion, heart, hepatopancreas, gill, muscle, and ovary were dissected out and stored at −80 °C. 

The ovarian samples were examined to assess the stage of ovarian development, based on previously published criteria [17]. Ovarian development can be divided approximately into five stages (Table 1) [17] and is assessed using the Gonad somatic index (GSI), calculated using Equation: GSI = gonadal weight/body weight × 100%.

The embryos and larvae were sampled according to the previous studies’ criteria. Embryonic and larval development can be divided approximately into different stages, as shown in Table 2 [17,18].

### 2.2. Nucleotide Sequence and Bioinformatics Analysis of Mn-CTSD

RNA was extracted from tissues of *M. nipponense* using RNAiso Plus reagents according to the manufacturer’s instructions (Takara, Japan). The first-strand cDNA was synthesized using the M-MLV reverse transcriptase kit according to the manufacturer’s instructions (Takara, Japan). Partial sequences of cathepsin D were obtained from the ovarian transcriptome library of *M. nipponense* (accessions SAMN11603268-SAMN11603282; www.ebi.ac.uk/biosamples/) (accessed on 5 May 2020). The complete open reading frame (ORF) of cathepsin D was cloned using a 3′RACE kit according to the manufacturer’s instructions [19]. The primers used are listed in Table 3.

DNAMAN 9.0 was used to analyze the amino acid sequence alignment between *Mn-CTSD* and CTSD from other species. The ORF Finder program (http://ncbi.nlm.nih.gov/gorf/gorf.html) (accessed on 19 May 2020) was used to predict the ORFs. BLASTX and BLASTN (http://www.ncbi.nlm.nih.gov/BLAST/) (accessed on 19 May 2020) were used to compare the sequence obtained with nucleotide and protein sequence analysis databases. Signal 3.0 (http://www.cbs.dtu.dk/services/SignalP/) was used for signal peptide prediction. Functional sites of amino acids were analyzed using PROSITE SCAN (http://npsa-pbil.ibcp.fr/cgi-bin/npsa_automat.pl?page=/NPSA/npsa_proscan.html) (accessed on 21 June 2020). A phylogenetic tree was constructed using MEGA5.0 software. *EIF* was used as the internal reference gene [20].

### 2.3. In Situ Hybridization

Ovaries of female *M. nipponense* were collected according to the requirements of the chromogenic ISH (CISH) technique and stored overnight in 4% paraformaldehyde phosphate buffer (PBS, pH 7.4) at 4 °C. ISH was performed based on the probe and CISH technique previously reported [21] The ovaries were dissected from mature female prawns and fixed in 4% paraformaldehyde in phosphate buffer saline (PBS, pH 7.4) at 4 °C overnight. ISH study was performed on 4 μm thick formalin fixed paraffin-embedded sections using Zytofast PLUS CISH implementation kit (Zyto Vision GmBH, Bremerhaven Germany). The blank control groups were routine hematoxylin-eosin staining (HE staining) sections without a probe poured and the negative control groups had antisense probes poured. The mRNA locations of *Mn-CTSD* were analyzed with a sense probe poured over slides. The anti-sense and sense probes in this study with digoxin signals were designed by Primer5 software based on the cDNA sequence of *Mn-CTSD* and synthesized by Shanghai Sangon Biotech Company (Table 3). All slides were examined under a light microscope for evaluation.

### 2.4. Mn-CTSD RNA Interference

Online software Snap Dragon (http://www.flyrnai.org/cgi-bin/RNAi_find_primers.pl) (accessed on 25 August 2020) was used to design primers that were then used for PCR amplification. After validation, the template was synthesized using a Transcript AidTMT7 High-yield transcription kit (Fermentas, MA, USA) using the *GFP* gene as a control. The purity and integrity of double-stranded (ds)RNA were determined by agarose gel electrophoresis. Concentrations of cathepsin D dsRNA were measured at 260 nm using a BioPhotometer (Eppendorf, Hamburg Germany) and stored at −80 °C until use (Table 3).

The experiment was carried out in October 2021 over a 30-day period. The prawns were randomly divided into either the experimental group (*n* = 60) or the control group (*n* = 60). The temperature of the water tank was kept at 24 ± 1 °C. Each prawn in the experimental group was injected with cathepsin D dsRNA, whereas those in the control group were injected with the same volume of GFP dsRNA. The injection site was the pericardial cavity, and the injection dose was 4 μg/g [18]. Injections were given once for 5 days [21,22]. On the 1st, 4th, and 7th day of injection, six prawns were randomly collected from each group and their ovarian tissues were dissected. The expression of cathepsin D was detected by qPCR using *EIF* as the internal reference gene [20]. During the experiment, the GSI was recorded to reflect the effect of cathepsin D dsRNA on ovarian maturation [17]. 

### 2.5. Statistical Analysis

The expression of cathepsin D was calculated by using the 2^−ΔΔCT^ method and analyzed using SPSS 23.0. Data from two groups were analyzed by one-way ANOVA and two-tailed Student’s *t*-tests. Differences were significant at the *p* < 0.05 level [23]. All quantitative fluorescence data are expressed as mean ± standard deviation. 

## 3. Results

### 3.1. Characteristics of Mn-CTSD Sequence

Full-length cDNA was obtained, with a total length of 2391 bp and containing an ORF of 1158 bp encoding 385 amino acids, and was named *Mn-CTSD* (GenBank accession no. ON220891). The ORF was verified by degenerate primers (Table 1). The molecular weight of *Mn-CTSD* was 41.96 kDa and its theoretical isoelectric point was 8.08. The amino acid sequence comprised: a hydrophobic signal peptide of 15 amino acids at the N terminus; a precursor domain of 24 amino acids; a mature domain of 332 amino acids; and an A tail-added signal (AATAA) and a poly(A) tail. Domain analysis showed that *Mn-CTSD* was a member of the aspartic protease family. *Mn-CTSD* contains two aspartic protease tags (VFDTGSSNLWV and AIADTGTSLIAA), one N-glycosylation site, and one proline ring spatial structure of the aspartic protease tag sequence (DVPPPMGP) (Figure 1).

### 3.2. Phylogenetic and Sequence Alignment Analysis of Mn-CTSD

Multiple sequence alignments were performed using DNAMAN 6.0 to compare *Mn-CTSD* with cathepsin D genes from other crustaceans and other animals. The results showed a similarity of *Mn-CTSD* with *CTSD* from *Macrobrachium rosenbergii*, *Palaemon carinicauda*, *P. japonicus*, *Penaeus monodon*, *Penaeus vannamei*, *Eriocheir sinensis*, *Procambarus clarkia,* and *Homarus americanus* of 95.54%, 91.45%, 87.31%, 86.79%, 85.75%, 84.42%, 82.60%, and 81.30%, respectively (Figure 2).

Compared with other invertebrates, the similarity of *Mn-CTSD* with *CTSD* from *Pinctada maxima*, *Pteria penguin*, *Callosobruchus maculatus*, *Bombus terrestris*, *Apis mellifera*, and *Polyrhachis vicina* was 51.03%, 64.54%, 71.79%, 69.21%, 68.99%, and 73.83%. Compared with vertebrates, the degree of similarity with *CTSD* from *Danio rerio*, *Lates calcarifer*, *Mus musculus*, and *Homo sapiens* was 57.39%, 57.93%, 53.28%, and 41.61%, respectively.

MEGA5.1 phylogenetic tree construction showed that *M. nipponense* clustered with *M. rosenbergii*, *P. monodon*, *E. sinensis* and other arthropods, and then with *B. terrestris*, *A. mellifera*, *P. maxima*, *P. penguin*, and other invertebrates, and finally with *D. rerio*, *M. musculus*, *H. sapiens*, and other vertebrates (Figure 3).

### 3.3. Spatiotemporal Expression of Mn-CTSD

*Mn-CTSD* was expressed at high levels in the eyestalk (E), cerebral ganglion (Cg), heart (H), hepatopancreas (He), gill (G), muscle (M), and ovary (O), with the expression in ovary being at least seven times higher than that in other tissues (*p* < 0.05). Its expression level was lowest in muscle (Figure 4A).

The relative expression level of *Mn-CTSD* was high in all five ovarian maturation stages in *M. nipponense*, being highest in O-IV (*p <* 0.05). There was a gradual increase in *Mn-CTSD* expression from stage O-I to stage O-III, followed by a sharp increase from stage O-III to O-IV, where it was more than two times that of other stages. Its expression decreased rapidly from O- to O-V, and there was no significant difference between O-V and O-I or between O-II and O-III (*p* > 0.05, Figure 4B). 

The relative expression levels of *Mn-CTSD* in different stages of embryo and larval development of *M. nipponense* were detected by qPCR. *Mn-CTSD* was expressed in all stages examined but was significantly higher in the cleavage stage (CS) (*p* < 0.05). During larval development, there was no significant difference in levels from L1 to L5, followed by a gradual increase from L5 to L15 (*p* < 0.05). From PL1 to PL15, expression of *Mn-CTSD* significantly increased (*p* < 0.05), peaking at PL5 (*p* < 0.05). From PL15 to PL25, its expression decreased gradually and reached the lowest value at PL25 (*p* < 0.05, Figure 4C).

### 3.4. Mn-CTSD Localization in Five Ovarian Stages by ISH

The position of *Mn-CTSD* in the ovary was located by ISH (Figure 5). The results showed *Mn-CTSD* signal in the nucleus, cell membrane, follicular membrane, and yolk granule of ovarian cells, being strongest in the nucleus. *Mn-CTSD* signal was detected in all five stages of ovarian maturation, being strongest in O-IV.

O-I (undeveloped stage, transparent), O-II (developing stage, yellow), O-III (nearly-ripe stage, light green), O-IV (ripe stage, dark green), O-V (worn out stage, gray). OC: oocyte; N: nucleus; CM: cytoplasmic membrane; Y: yolk granule; FC: follicle cell; FM: follicle membrane; Scale bars: Low magnification 100×, High magnification 400×. HE represents the blank control groups with routine hematoxylin-eosin staining. Negative represents the control groups with antisense probes poured. Positive represents the experimental group with sense probes poured.

### 3.5. Effect of Mn-CTSD on Ovarian Maturation Was Knocked Out by RNAi

Samples of two groups were taken on Days 1, 4, and 7 after injection. qPCR results showed that compared with the control group, the expression level of *Mn-CTSD* in the experimental group decreased significantly on Day 4 and Day 7 of injections. On day 4, the expression level of the experimental group decreased by 98.80% compared to the control group. On Day 7, the expression level of the experimental group decreased by 99.48% compared to the control group (*p* < 0.01, Figure 6). 

The effect of *Mn-CTSD* on ovarian maturation was studied by observing gonadal development. On Day 1 of injection, the GSI was 1.24% in the control group and 1.00% in the experimental group. Both the control group and the experimental group were in stage I of ovarian maturation. By Day 10, significant differences in the GSI, being 2.37% in the control group and 1.45% in the experimental group (*p* < 0.05). By Day 20, the average GSI of the control and the experiment groups were significantly different (3.19% versus 1.64%, respectively; *p* < 0.01). By Day 30, the GSI of the control group was 4.57% versus 1.79% in the experimental group (*p* < 0.01). Ovary development reached O-III (nearly-ripe stage, light green) in 80% of the control group versus 0% in the experimental group, with only 86.5% of ovaries in the experimental group reaching O-II (developing stage, yellow). Thus, the ovarian maturation rate of the experimental group was significantly slower than that of the control group (Figure 7).

## 4. Discussion

The full-length sequence of *Mn-CTSD* was obtained from the ovarian transcriptome library of *M. nipponense*. Similar to that in other species, the sequence also comprised three parts: signal peptide, precursor domain, and mature domain [24,25]. There was a characteristic sequence (DVPPPMGP), which occurred as a proline ring; this characteristic sequence only occurs in nondigestive cathepsin D [26]; thus, *Mn-CTSD* is a nondigestive cathepsin. The *Mn-CTSD* sequence showed more than 80% similarity to *M. rosenbergii*, *P. carinicauda*, *P. japonicus*, *P. monodon* and other crustaceans. The *Mn-CTSD* sequence showed more than 50% similarity to *P. maxima*, *P. penguin*, *C. maculatus*, *B. terrestris*, *A. mellifera*, and other invertebrates, whereas its level of similarity with zebrafish, humans, and other vertebrates was more than 40%, indicating that it has a relatively conserved structure. The phylogenetic relationship of the *Mn-CTSD* amino acid sequence was consistent with that of a traditional classification. There are three *N*-glycosylation sites in *Sus scrofa* [27], two in *Exopalaemon carinicauda*, and one in *M. nipponense*. Glycosylation is an important post-translational modification of proteins and has an important role in many biological processes, such as cellular immunity, protein translation regulation, and protein degradation [7]. Thus, the difference in the numbers of glycosylation sites might be related to their function during glycosylation. 

Cathepsin D is a lysosomal aspartate endonuclease originally identified as a housekeeping gene for the removal of excess proteins. It is expressed in almost all eukaryotic cells and has an important role in metabolism and other activities in vivo [28,29,30,31]. The expression analysis of *Mn-CTSD* in different tissues of *M. nipponense* showed that it was expressed in all tissues, suggesting that cathepsin D is involved in various physiological activities in *M. nipponense*. Although the highest expression level of cathepsin D was found in the gonads of grass carp (bladder, intestine, liver, heart, muscle, brain, spleen, fin, gill, head kidney, gonad, and eye) [12] research on its role in reproduction has been lacking. Instead, research on cathepsin D has mainly focused on its role in immunity [27,32,33,34]. In the current study, among the selected tissues of *M. nipponense* examined (muscles, heart, ovary, cerebral ganglion, eyestalks, hepatopancreas, and gills), *Mn-CTSD* expression was highest in the ovary, suggesting that it has a role in this tissue. During ovarian maturation, expression of *Mn-CTSD* peaked in O-IV, verifying the correlation between cathepsin D and ovarian maturity. Given the levels of *Mn-CTSD* expression in different stages of embryonic development combined with the expression of different stages of ovarian maturation, we speculate that *Mn-CTSD* is a maternal gene.

Cathepsin D might be involved in embryo development, and its expression varies in different species. It is also implicated in egg development and tissue invasion [35]. During embryonic development in *Xenopus laevis*, when cathepsin D activity occurs in all embryonic development stages from 0 to 40, gradually increasing from stage 0 to 23 [36]. The expression level and enzyme activity of cathepsin D decreased gradually during embryo development of *Seriola lalandi* [37], whereas, during embryonic development of *Ctenopharyngodon idella*, it showed a gradual increase [12]. The expression level of cathepsin D in *Oncorhynchus mykiss* was higher in the E1 and E2 stages, lower in the E3 stage, and lower still and stable in the E4–E9 stages. Its expression was consistently high during the early embryo stages and then plateaued during the late embryo stages [38]. The results of ISH showed that the signal was strongest at O-IV, which was consistent with the expression of *Mn-CTSD* at different stages of ovarian maturation, confirming the correlation between the expression of cathepsin D and ovary maturation.

Cathepsin D is an important protease that is involved in a variety of physiological and pathological functions. The dynamic balance between high expression of cathepsin D and cathepsin D inhibitors is disrupted, leading to the transformation of macrophages into foam cells, which are prone to apoptosis mediated by cathepsin D, leading to atherosclerosis [39,40]. Follo et al. reported that cathepsin D could relieve muscle atrophy in zebrafish by knocking out the cathepsin D gene [35]. Cathepsin D also has an important role in the immune system. When Bonavida et al. investigated the function of Cathepsin D, they found that it was not only involved in the activation of secreted proteins but also closely related to immune processes, such as antigen delivery [41]. In studies of RNAi knockdown of cathepsin D in *P. clarkii*, it was demonstrated that cathepsin D has an important immune function in this organism [14].

Through a 30-day RNAi experiment, the GSI in the group injected with ds*Mn-CTSD* was significantly lower than that in the control group. By Day 30, the ovarian maturation of the experimental group remained at O-I, with little change from the gonadal development of Day 1, whereas the ovarian maturation of the control group reached O-III, and gonadal development was close to maturity. Thus, ds*Mn-CTSD* injection had an inhibitory effect on ovarian maturation, confirming the important role of cathepsin D in this process.

## 5. Conclusions

In this study, the full-length cDNA sequence of *Mn-CTSD* from *M. nipponense* was successfully cloned and analyzed. The expression characteristics of cathepsin D in different tissues, and ovary and embryo developmental stages of *M. nipponense* were studied. RNAi experiments confirmed that *Mn-CTSD* has an important role in ovarian maturation. Thus, these results provide a theoretical basis for further research on the molecular mechanism of ovarian maturation and a new approach for solving the problem of rapid sexual maturation in this economically important species.

## Figures and Tables

**Figure 1 genes-13-01495-f001:**
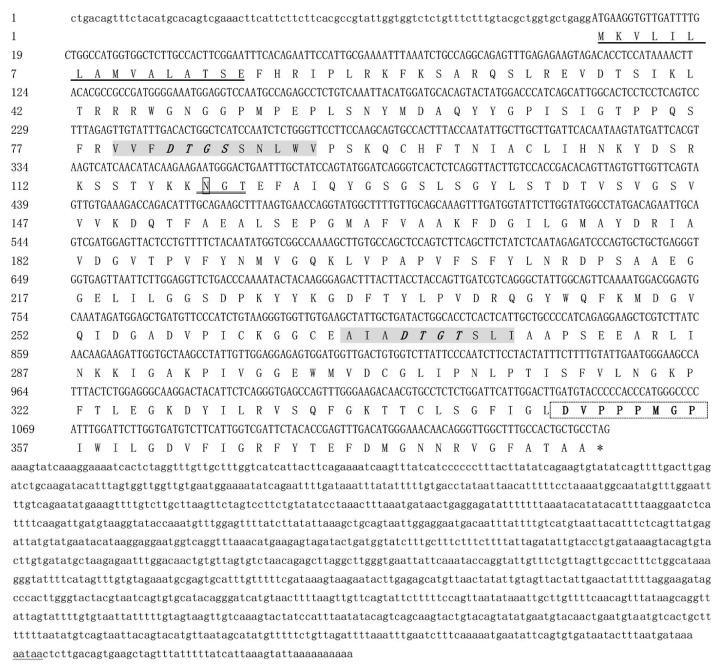
Full cDNA sequence and predicted amino acid sequence of *Mn-CTSD.* Signal peptides are represented by a solid underline; The N-glycosylation sites were in the square box of thick line. The two aspartic protease signature sequences were marked with gray shadows. The catalytic motifs (-DTGS-, -DTGT-) were in bold italics, and the enzyme activation sites were in bold and single underline. Non-digestive *Mn-CTSD* signature sequences are shown in bold with dotted boxes. Wavy line labeling is a tail signal.

**Figure 2 genes-13-01495-f002:**
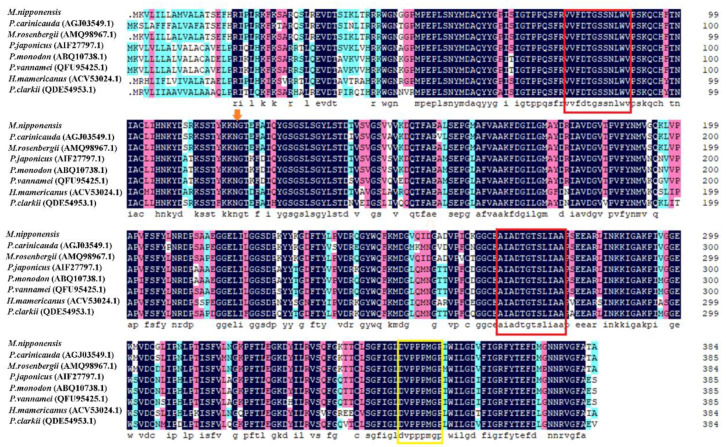
Comparison of *Mn-CTSD* derived amino acid sequences with cathepsin D from other crustaceans. Sequence uniformity is shown in black, with pink indicating conservative variation. N-glycosylation sites are marked by red arrows, aspartic protease signature sequences are marked by red boxes, and non-digestive cathepsin D signature sequences are marked by yellow boxes.

**Figure 3 genes-13-01495-f003:**
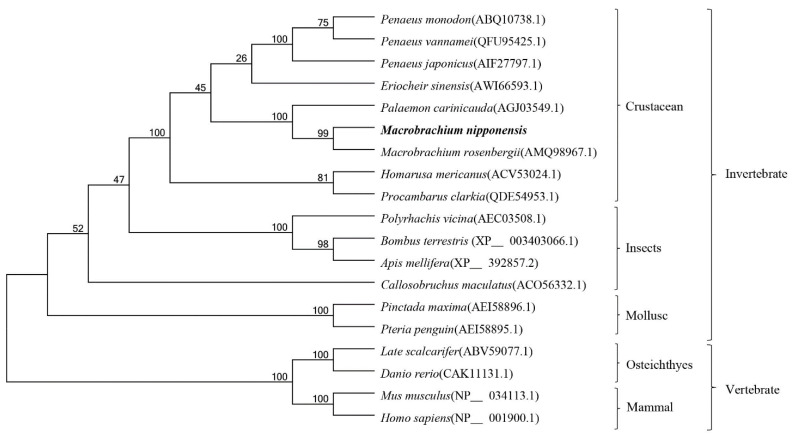
Phylogenetic tree of *Mn-CTSD*. The graph was generated by the MEGA5.0 program using the adjacency method. Bootstrap copy to 1000. The GenBank login number is in square brackets.

**Figure 4 genes-13-01495-f004:**
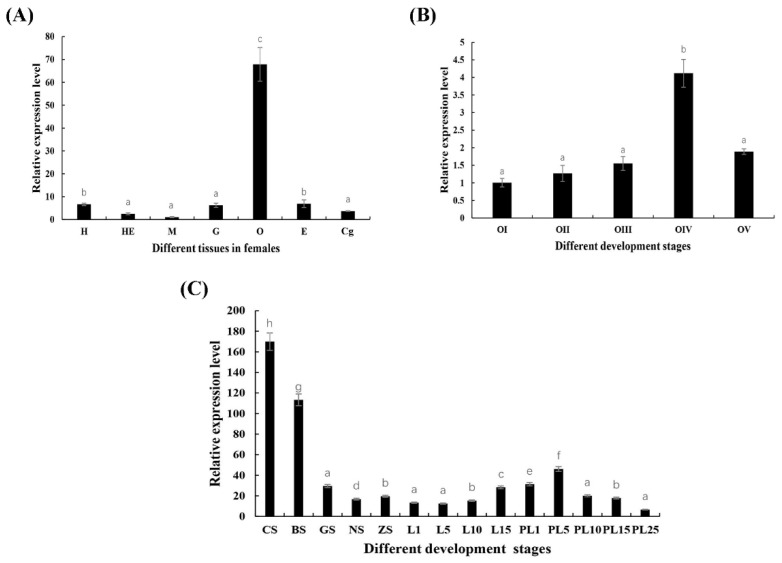
The expression pattern of the *Mn-CTSD* gene in different tissues (**A**) and different stages of ovarian maturation (**B**) and developmental stages (**C**) of *M. nipponense* were measured by qPCR. E—eye, Cg—cerebral ganglion, H—heart, He—hepatopancreas, G—gill, M—muscle, O—ovary; Data are presented as the mean ± SD (*n* = 6). *p* < 0.05 was considered to be statistically significant. significant a, b, c, d, e, f indicate significant differences.

**Figure 5 genes-13-01495-f005:**
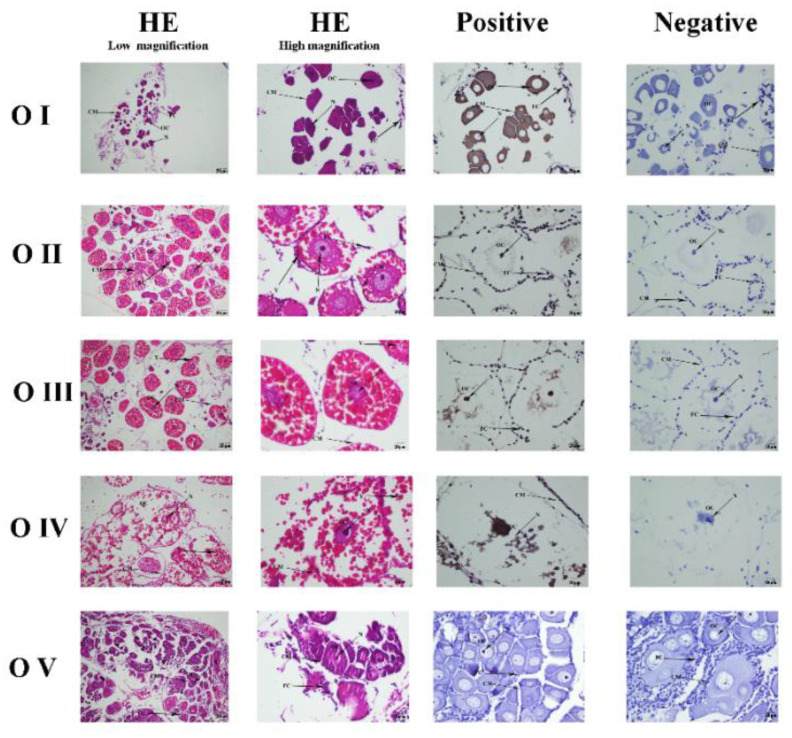
Representative histological sections showing *Mn-CTSD* expression during different ovarian maturational stages.

**Figure 6 genes-13-01495-f006:**
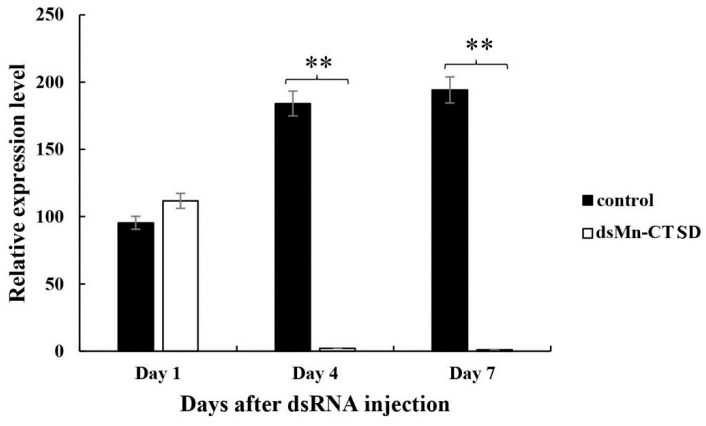
Expression levels of *Mn-CTSD* in ovaries after RNA interference. Data are shown as mean ± standard deviation (n = 6). Using the paired sample T test, asterisks indicate significant differences (*p* < 0.05). ** denotes statistical significance of *p* < 0.01.

**Figure 7 genes-13-01495-f007:**
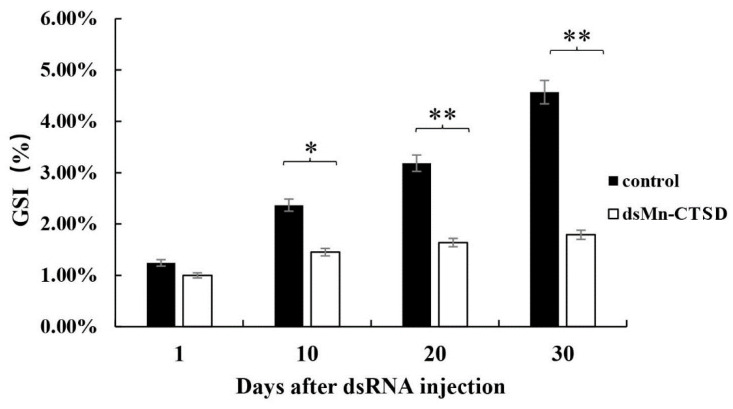
GSI (%) of female *M. nipponense* after ds *Mn-CTSD* injection. Data are shown as mean ± standard deviation (*n* = 6). Using one-way ANOVA, letters indicated significant differences (*p* < 0.05). * denotes statistical significance of *p* < 0.05. ** denotes statistical significance of *p* < 0.01.

**Table 1 genes-13-01495-t001:** Different ovarian stages of *M. nipponense*.

Ovarian Stage	Ovarian Maturation Stage	Ovarian Appearance Color
O-I	undeveloped stage	transparent
O-II	developing stage	yellow
O-III	nearly-ripe stage	light green
O-IV	ripe stage	dark green
O-V	worn out stage	gray

**Table 2 genes-13-01495-t002:** Different embryonic and larva development stages of *M. nipponense*.

Different Embryonic and Larva Development Stages	
CS	cleavage stage
BS	blastula stage
GS	gastrulation stage
NS	nauplius stage
ZS	zoea stage
L1	the 1st day larvae after hatching
L5	the 5th day larvae after hatching
L10	the 10th day larvae after hatching
L15	the 15th day after hatching
PL1	the 1st day after metamorphosis
PL5	the 5th day after metamorphosis
PL10	the 10th day after metamorphosis
PL15	the 15th day after metamorphosis
PL25	the 25th day after metamorphosis

**Table 3 genes-13-01495-t003:** Primers involved in this study.

Primer Name	Sequence (5′-3′)	Usage
*Mn-CTSD 3′F1*	TGAAGGTGTTGATTTTGCTGGC	3′RACE
*Mn-CTSD* 3′F2	CTCCATAAAACTTACACGCCGC	3′RACE
*Mn-CTSD* 3′F3	TCATCCAATCTCTGGGTTCCTTC	3′RACE
*Mn-CTSD* qF	GACGTTGTCTTCCCAAACTGG	RT-PCR
*Mn-CTSD* qR	GCTTGGAGGTTCTGACCCAAA	RT-PCR
EIF-F	CATGGATGTACCTGTGGTGAAAC	RT-PCR
EIF-R	CTGTCAGCAGAAGGTCCTCATTA	RT-PCR
*Mn-CTSD* iF	TAATACGACTCACTATAGGGGACGTTGTCTTCCCAAACTGG	*CTSD* dsRNA F
*Mn-CTSD* iR	TAATACGACTCACTATAGGGGCTTGGAGGTTCTGACCCAAA	*CTSD* dsRNA R
*GFP* iF	GATCACTAATACGACTCACTATAGGGTCCTGGTCGAGCTGGACGG	*GFP* F
*GFP* iR	GATCACTAATACGACTCACTATAGGGCGCTTCTCGTTGGGGTCTTTG	*GFP* R
probe	CTCGTCTTATCAACAAGAAGATTGGTGCTAAGCCTATTGTTGGAGGAGAGTGGATGGTTGACTGTGGTCTTA	ISH
anti-probe	TAAGACCACAGTCAACCATCCACTCTCCTCCAACAATAGGCTTAGCACCAATCTTCTTGTTGATAAGACGAG	ISH

## Data Availability

Data is contained within the article. The data presented in this study are available in [insert article here].

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
