# Peer review of "Cathepsin D Plays a Vital Role in Macrobrachium nipponense of Ovary Maturation: Identification, Characterization, and Function Analysis"

_genes, 2022, doi:10.3390/genes13081495_

Round 1

Reviewer 1 Report

Manuscript Number: Genes 1814334

 Title: Cathepsin D plays a vital role in Macrobrachium nipponense of ovary maturation: identification, characterization and function analysis.

 General comments:

The manuscript deals with the molecular characterization of the Cathepsin D gene in the freshwater prawn Macrobrachium nipponense, which is an important species for Chinese aquaculture. This species undergoes a fast sexual maturity, thus becoming a disadvantage for grow-out purposes. Therefore, this study focused on the role of cathepsin D on ovary maturation and assayed a RNAi approach to determine its role.

The complete gene and open reading frame of cathepsin D was obtained and sequenced. Then, primers for qPCR, in situ hybridization (ISH) and RNAi were designed in order to study the gene localization, quantity and effect on ovary maturation, respectively.

The total length of the gene was 2391 bp and consisted of an open reading frame (ORF) of 1158 bp encoding 385 amino acids, with presence of conserved N-glycosylation sites and sequences of nondigestive cathepsin D. Assays using qPCR showed that cathepsin D is expressed in various tissues, but predominantly in ovary. Assay using ISH showed that cathepsin D is increasingly present in ovary from stage O-II and O-III, but with the highest presence in stage O-IV, then almost absent in stage O-V. A RNAi experiment was done to determine whether silencing the cathepsin D gene had an effect in ovary maturation. Results showed that silencing cathepsin D gene significantly reduced ovary maturation in experimental animals compared to controls. Authors conclude that cathepsin D gene has an important role in ovarian maturation, and that these results lay the theoretical basis for further research of the molecular mechanism of ovarian maturation. They propose the use of RNAi as a new approach for solving the issue of rapid sexual maturation in this economically important species.

The manuscript is interesting and the results are compelling on the importance of such gene on ovary maturation in this species. The manuscript is in general well written in English, but it is advised to review it again in order to improve some grammatical or spelling typos. Some specific comments are given below:

 Specific comments:

 Abstract

 Comment 1 - Lines 14-15. Please mention one or two of the main drawbacks of the fast sexual maturation of female M. nipponense.

Comment 2 - line 19 should say: “The qPCR analysis…”

Comment 3 – line 20. Delete the word “so”

Comment 4 – line 23. Please include the ovarian developmental stage or age/size at which female prawns were treated with RNAi.

Introduction

 Comment 5 – line 57. The species should be Procambarus clarkii

Comment 6 – line 58. Is the genus Macrobrachium correct? Isn´t it Marsupenaeus? Also, the corrected genus of penaeid shrimp is at present Penaeus. The taxonomic proposal of Perez-Farfante and Kensley has been shown to be incorrect. It is advised to correct the genus names of penaeid shrimp species to Penaeus.

Comment 7 – lines 60 - 66. It is advised to re-write this paragraph to improve clarity. The paragraph should clearly state the aims or objectives of the study. As it presently stands, this paragraph seems to be part of the methodology.

 Material and methods

 Comment 8 – line 78. Please state the temperature at which the samples were stored.

Comment 9 – line 81. Please change the word “staging” for the word “development”

Comment 10 – Table 1. Line O-V change the word “spent” for the words “worn out”.

Comment 11 – line 88. Please delete the words “Based on previous studies”.

Comment 12 – line 131 should say: “…were injected with the same volume of GFP dsRNA”

Comment 13 – line 133. Please state how many times the RNAi injections were administered at 5-day intervals. If it was administered only once for 5 days, please re-write the sentence to improve clarity.

Results and discussion

 Comment 14 – line 175. I think a sentence should be added stating the similarity of cathepsin D gene compared to other crustacean species as well. Later on it is said and shown that cathepsin D from M. nipponense cluster with that of other crustaceans, the similarity percentage is not shown. So, it should be stated in this paragraph.

Comment 15 – lines 222 - 225. These sentences are confusing. The figure 6 shows that at day 4 and 7 a significant reduction of cathepsin D expression occurred in the RNAi treated group compared to controls, but this sentence seems to compare the expression between treatment at day 4 with that of day 7. Authors should make sure what expression are you trying to compare, between treatment and control, or the expression reduction in treatment at days 4 and 7? Please correct this sentence accordingly.

Comment 16 – lines 234 – 235 should say: “By Day 10, significant differences in GSI were found, being 2.37% in the control group and 1.45% in the experimental group…”

Comment 17 – lines 254-257. This sentence refers again to the similarity between cathepsin D genes in invertebrates and vertebrates, but no reference or comparison is done to the genes of other crustaceans. Please include the similarity comparison with crustaceans.

Comment 18 – line 312. It seems that the word “experimental” refers instead to the control group, isn’t it? Please make sure the sentence refers to the experimental group or to the control group.

Author Response

Dear Reviewer ,

Thank you very much for your comments and suggestions. The valuable comments from you not only helped us with the improvement of our manuscript, but suggested some ideas for future studies.

Below you will find our responses to your comments:

Comment 1- Lines 14-15. Please mention one or two of the main drawbacks of the fast sexual maturation of female M. nipponense.

Response: The fast sexual maturation produced a large number of offspring which lead to the reduction of resilience and survival rate and increase of hypoxia risk, and then seriously affect the economic benefits of prawn farming.

Comment 2- line 19 should say: “The qPCR analysis…”

Response: Thank you for your correction. The mistake has been corrected.

Comment 3– line 20. Delete the word “so”.

Response: Thank you for your correction. The mistake has been corrected

Comment 4 line 23. Please include the ovarian developmental stage or age/size at which female prawns were treated with RNAi.

Response: Thank you very much for your suggestion, we have revised this part in the revised manuscript.

Comment 5– line 57. The species should be Procambarus clarkii

Response: Thank you for your correction. The mistake has been corrected.

Comment 6– line 58. Is the genus Macrobrachium correct? Isn´t it Marsupenaeus? Also, the corrected genus of penaeid shrimp is at present Penaeus. The taxonomic proposal of Perez-Farfante and Kensley has been shown to be incorrect. It is advised to correct the genus names of penaeid shrimp species to Penaeus.

Response: Thank you for your correction. The mistake has been corrected.

Comment 7– lines 60 - 66. It is advised to re-write this paragraph to improve clarity. The paragraph should clearly state the aims or objectives of the study. As it presently stands, this paragraph seems to be part of the methodology.

Response: Thank you very much for your suggestion. We rewrite this paragraph in the revised manuscript.

Material and methods

Comment 8 –line 78. Please state the temperature at which the samples were stored.

Response: The prawns were then euthanized and tissues including the eyestalk, cerebral ganglion, heart, hepatopancreas, gill, muscle, and ovary were dissected out and stored at -80℃.

Comment 9 line 81. Please change the word “staging” for the word “development”

Response: Thank you for your correction. The mistake has been corrected.

Comment 10 Table 1. Line O-V change the word “spent” for the words “worn out”.

Response: Thank you for your suggestion. We have changed the word according to your suggestion.

Comment 11line 88. Please delete the words “Based on previous studies”.

Response: We delete the words “Based on previous studies”.

Comment 12 line 131 should say: “…were injected with the same volume of GFP dsRNA”

Response: We here changed to “…were injected with the same volume of GFP dsRNA”.

Comment 13line 133. Please state how many times the RNAi injections were administered at 5-day intervals. If it was administered only once for 5 days, please re-write the sentence to improve clarity.

Response: We re-write the sentence the word according to your suggestion.

Results and discussion

Comment 14line 175. I think a sentence should be added stating the similarity of cathepsin D gene compared to other crustacean species as well. Later on it is said and shown that cathepsin D from M. nipponense cluster with that of other crustaceans, the similarity percentage is not shown. So, it should be stated in this paragraph.

Response: Thank you for your suggestion. We have added more details in this paragraph.

Comment 15lines 222 - 225. These sentences are confusing. The figure 6 shows that at day 4 and 7 a significant reduction of cathepsin D expression occurred in the RNAi treated group compared to controls, but this sentence seems to compare the expression between treatment at day 4 with that of day 7. Authors should make sure what expression are you trying to compare, between treatment and control, or the expression reduction in treatment at days 4 and 7? Please correct this sentence accordingly.

Response: Here we corrected to “On day 4, the expression level of the experimental group decreased by 98.80% compared to the control group. On Day 7, the expression level of the experimental group decreased by 99.48% compared to the control group.”

Comment 16lines 234 – 235 should say: “By Day 10, significant differences in GSI were found, being 2.37% in the control group and 1.45% in the experimental group…”

Response: We corrected this part in the manuscript according to your suggestion.

Comment 17lines 254-257. This sentence refers again to the similarity between cathepsin D genes in invertebrates and vertebrates, but no reference or comparison is done to the genes of other crustaceans. Please include the similarity comparison with crustaceans.

Response: Thanks for your suggestion, and now the relevant description has been revised.

Comment 18line 312. It seems that the word “experimental” refers instead to the control group, isn’t it? Please make sure the sentence refers to the experimental group or to the control group.

Response: Thanks for your suggestion, and we corrected the mistake.

Reviewer 2 Report

Comments on the manuscript:

“Cathepsin D plays a vital role in Macrobrachium nipponense of ovary maturation: identification, characterization and function analysis”

The Macrobrachium nipponense river shrimp is an economically important aquaculture species. But his breeding is hampered by the rapid sexual maturation of the females. In the present study, the cDNA of the gene encoding cathepsin D, a lysosomal protein involved in ovarian maturation, was cloned from M. nipponense. PCR analysis indicated that this gene was expressed in all the tissues tested, and most strongly in ovaries. In situ hybridization has shown its localization mainly in the nuclei of oocytes. During development, protein expression is maximal at stage O-IV of ovarian maturation. Injection of interfering RNA is manifested by a decrease in the expression of the gene encoding cathepsin in the ovaries, and the somatic gonado-somatic index of these animals is significantly lower than that of animals not having received such an injection. Ovarian development reached stage O-III in 80% of animals not injected and 0% in animals injected with this interferent RNA.

This study is interesting and needs some improvements to the manuscript. Here are some remarks.

Page 1, abstract and introduction: use italics for Macrobrachium nipponense and M. nipponense. The same applies for Procambarus clarkia, Fenneropenaeus chinensis.

Page 2, line 57: the reference “Yang et al., 2008” is not in the list.

Page 2, line 60: In the introduction, the method used is well explained, but I did not find the purpose of the study. It would be useful to find here the purpose of the study clearly.

Page 2, material and method, line 82: the reference “Qiao et al., 2019” is not in the reference list.

Page3, table 2. Is it “the 5th day larvae after hatching” instead of “the 5th larvae after hatching”?

Is it also “the 15th day larvae after hatching” instead of “the 15th larvae after hatching”?

Page 4, line 114: give some explanation about the in situ hybridization technique (even if it has been previously reported).

Page 4, line 134: don’t use italics to write “prawns”.

Page 9, figure 5. This figure needs to be completed.

- scale bars are not enough visible on the pictures.

- captions are not visible on the sections.

- Indicate in the legend the meaning of O-I, O-II, O-III, O-IV and O-V.

- what were the stainings used? I suppose HE is hemalum-eosin; but what are “positive” and “negative”? There is no indication in Materials and Methods where a paragraph on staining should be added.

Check references carefully and write them in journal style. I noticed a few typos.

Page 13, line 376: What is “Fa?Anha A.R.,”? “Façanha43, I suppose.

Page 14, reference 16: this reference is not complete.

Page 14, reference 17: this reference is not complete.

Page 15, reference 33: correct “Roberg K., K?Gedal K., ?Llinger K”.

Page 15, reference 41: this reference is not complete.

Author Response

Dear Reviewer,

Thank you very much for your comments and suggestions. The valuable comments from you not only helped us with the improvement of our manuscript, but suggested some ideas for future studies.

Below please will find our responses to your comments:

1.Page 1, abstract and introduction: use italics for Macrobrachium nipponense and M. nipponense. The same applies for Procambarus clarkia, Fenneropenaeus chinensis.

Response: Thank you for your correction, we have corrected in the revised manuscript.

2.Page 2, line 57: the reference “Yang et al., 2008” is not in the list.

Response: We add this reference in the list.

3.Page 2, line 60: In the introduction, the method used is well explained, but I did not find the purpose of the study. It would be useful to find here the purpose of the study clearly.

Response: Thank you for your suggestion. We rewrite this paragraph and make the purpose clearly.

4.Page 2, material and method, line 82: the reference “Qiao et al., 2019” is not in the reference list.

Response: Thanks for the reminder, we add this reference in the list.

5.Page3, table 2. Is it “the 5th day larvae after hatching” instead of “the 5th larvae after hatching”?Is it also “the 15th day larvae after hatching” instead of “the 15th larvae after hatching”?

Response: I have corrected these two mistakes.

6.Page 4, line 114: give some explanation about the in situ hybridization technique (even if it has been previously reported).

Response: Thanks for your suggestion, now the details about ISH has been added.

7.Page 4, line 134: don’t use italics to write “prawns”.

Response: We have corrected this mistake.

8.Page 9, figure 5. This figure needs to be completed.

- scale bars are not enough visible on the pictures.

- captions are not visible on the sections.

- Indicate in the legend the meaning of O-I, O-II, O-III, O-IV and O-V.

- what were the stainings used? I suppose HE is hemalum-eosin; but what are “positive” and “negative”? There is no indication in Materials and Methods where a paragraph on staining should be added.

Response: Thank you very much for your advice. We have corrected all the above problems. We have made the scale bars more clearly in the revised figure 5. The meaning of O-I, O-II, O-III, O-IV and O-V are indicated in legend. HE represents the blank control groups with routine hematoxylin-eosin staining. Negative rep-resents the control groups with antisense probes poured. Positive represents the experimental group with sense probes poured.

10.Check references carefully and write them in journal style. I noticed a few typos.

Page 13, line 376: What is “Fa?Anha A.R.,”? “Façanha43, I suppose.

Page 14, reference 16: this reference is not complete.

Page 14, reference 17: this reference is not complete.

Page 15, reference 33: correct “Roberg K., K?Gedal K., ?Llinger K”.

Page 15, reference 41: this reference is not complete.

Response: Thank you for your correction. I have revised the relevant references.